# Generalized Lindblad Master Equation for Measurement-Induced Phase Transition

Yi-Neng Zhou

Institute for Advanced Study, Tsinghua University, Beijing,100084, China
* zhou-yn19@mails.tsinghua.edu.cn

November 12, 2022

## Abstract

The measurement-induced phase transition (MIPT) occurs when the system is evolving under unitary evolution together with local measurements followed by post-selection. We propose a generalized version of the Lindblad master equation as a continuous equation, to describe the dynamics of the second Rényi entropy in the MIPT. This generalized Lindblad equation explicitly takes into account the post-selection in the MIPT, which is realized as the Einstein-Podolsky-Rosen (EPR) state projection in the equation. Also, this generalized Lindblad equation preserves the Hermitian, unit trace, and positive definiteness of the density matrix. We further use the hard-core Bose-Hubbard model as a concrete example to numerically confirm that our generalized Lindblad equation is applicable to describing the MIPT.

# 1   Introduction

Recent years have seen major progress in the understanding of quantum entanglement, and the study of entanglement transitions that separate different entanglement phases have been wildly discussed. Since the unitary evolution of a closed system typically drives it towards volume-law scaling for the entanglement entropy of subsystems, adding local measurement in the evolution process has been raised as a method to restrict the growth of entanglement. The study of the entanglement transition gave rise to the concept of the measurement-induced phase transition (MIPT). In measurement-induced phase transition, the unitary evolution can establish the entanglement between different parts of the system, while the local projective measurements on the system are believed to destroy the entanglement between different parts of the system. Thus, there is a competing relation between the unitary evolution and local measurements, and therefore, the increase in measurement rate can lead to an entanglement phase transition [1–7]. This dynamical process in some cases can be mapped to a classical percolation problem, making it feasible for large-size classical simulation [8]. It also enables the study of the phase diagram and critical exponents in different regions [9–15]. The nature of this phase transition has been analyzed from different perspectives including classical statistical mechanics models [8, 16], information scrambling [17], quantum error corrections [18, 19] and symmetry breaking [20].

The methods of studying entropy dynamics under measurements include tensor network [21–23], matrix product state [24], random unitary circuit [16, 25]. However, it remains an open question to find an equation that is continuous in time to describe the entropy dynamics of a quantum system along this unitary evolution together with measurements. Also, the role of post-selection is essential in this entanglement phase transition since it excludes the entropy corresponding to the probability distribution of different measurement results, and thus makes it possible to obtain the entanglement transition. Therefore, it should be explicitly written in this continuity equation. Here, we consider the dynamics of the second Rényi entropy of the system under unitary evolution together with projective measurements, followed by post-selections to project a general mixed state to a pure state.

In this paper, we derive a dynamical equation of density matrix to describe the second Rényi entropy dynamics under unitary evolution and projective measurements that are followed by post-selections. This generalized Lindblad equation preserves the Hermitian, unit trace, and positive definiteness of the density matrix. Moreover, in this process, the entropy comes from two parts: the entanglement entropy of the system corresponding to different measurement results, and the entropy corresponding to the probability distribution of different measurement results. In the MIPT, we only care about the entropy of the former part, so we need to exclude the entropy of the latter part. In our equation, we use the measurement on a partial basis to directly exclude this part of

entropy. Since the measurement on the partial basis is explicitly written in our equation, we can observe the entanglement phase transition by directly calculating the entanglement entropy from our equation. Thus, we do not need to run the same protocol many times and post-process the entropy results to observe this phase transition. Our generalized Lindblad equation thus provides a natural description of the entropy dynamics along this process.

## 2  The generalized Lindblad equation for measurement process

We first briefly review how to obtain a Lindblad-like equation that describes a system under unitary evolution together with frequent measurements [26–29].

For a closed system under unitary evolution, the time evolution of the density matrix of the system follows:

$$\frac{\partial \rho}{\partial t} = -i[\hat{H}, \rho]. \tag{1}$$

Thus, to the first order of $\delta t$, we obtain

$$\rho(t + \delta t) \simeq \rho(t) - i[\hat{H}, \rho(t)]\delta t + o(\delta t)^2. \tag{2}$$

Next, we consider the system being measured at an equal time interval $\delta t$. During two neighboring measurements, the system's evolution is governed by $\hat{H}$. Then, the density matrix after the measurements is

$$\rho^M(t + \delta t) = \sum_a \hat{L}_a \rho(t + \delta t)\hat{L}_a^\dagger. \tag{3}$$

Here, we use $\rho^M$ to denote the density matrix after measurements ($M$), and $\hat{L}_a$ is called the Lindblad operator or quantum jump operator. Here, $a = 1, ..., n$ labels the possible quantum jumps resulting from measurements.

If we simply assume that the probability of the system being measured at time $t+\delta t$ is $P(t+\delta t)$, then the density matrix after this probabilistic measurement is:

$$\rho^M(t + \delta t) = [1 - P(t + \delta t)]\rho(t + \delta t) + P(t + \delta t)\sum_a \hat{L}_a \rho(t + \delta t)\hat{L}_a^\dagger. \tag{4}$$

Here, we consider the system is projected to a complete basis

$$\sum_{a=1}^{n} \hat{L}_a^\dagger \hat{L}_a = \mathcal{I}, \tag{5}$$

and this assumption preserves the trace of density matrix along this measurement process. By assuming the completeness of measurement basis and taking the limit $\delta t \to 0$ then we obtain a differential equation of density matrix

$$\frac{\partial \rho}{\partial t} = -i[H, \rho(t)] + \eta(t)\sum_{a=1}^{n}\left[\hat{L}_a \rho(t)\hat{L}_a^\dagger - \frac{1}{2}\{\hat{L}_a^\dagger \hat{L}_a, \rho(t)\}\right]. \tag{6}$$

Here, $\eta(t)$ is the probability of the system being measured per unit time, and it is defined as $\eta(t + \delta t) = \frac{P(t+\delta t)}{\delta t}$. Here, we suppose that there is no singularity in $\eta(t)$. This equation is the same as the Lindblad master equation if we regard the measurement rate $\eta(t)$ as the dissipation strength $\gamma$.

# 3 The measurement process followed by post-selection

When one considers the case where the system is projected on a partial basis, the Eq. (6) is not able to describe the density matrix dynamics. Here, projection on a partial basis means we only save the result of the system being projected on some specific states $b$ where $b = 1, 2, ..., m$ and $m < n$. To see the failure of the Eq. (6) in this case, we can first directly use the Eq. (6) by changing the summation on the right-hand side from $a = 1, ..., n$ to $a = 1, ..., m$. Then, we will find that this new equation does not preserve the trace of the density matrix. Therefore, to preserve the unit trace of the density matrix, we need to normalize the density matrix when the system has been measured. Thus, in comparison to the Eq. (4), after this probabilistic measurement and the followed post-selection, the density matrix becomes:

$$\rho^M(t + \delta t) = [1 - P(t + \delta t)]\rho(t + \delta t) + P(t + \delta t)\frac{\sum_{a=1}^m \hat{L}_a\rho(t + \delta t)\hat{L}_a^\dagger}{\mathrm{Tr}\left(\sum_{b=1}^m \hat{L}_b\rho(t)\hat{L}_b^\dagger\right)}. \tag{7}$$

Here, the denominator on the right-hand side is the normalization factor due to post-selection. Similar to the Eq. (6), by taking the limit $\delta t \to 0$, we further obtain:

$$\frac{\partial \rho}{\partial t} = -i[\hat{H}, \rho(t)] + \eta(t)\sum_{a=1}^m \frac{\hat{L}_a\rho(t)\hat{L}_a^\dagger}{\mathrm{Tr}\left(\sum_{b=1}^m \hat{L}_b\rho(t)\hat{L}_b^\dagger\right)} - \frac{\eta(t)}{2}\sum_{a=1}^n \{\hat{L}_a^\dagger\hat{L}_a, \rho(t)\}. \tag{8}$$

If we set $m = n$ which means we measure the system on a complete basis, we will find that the normalization factor $\mathrm{Tr}\left(\sum_{b=1}^n \hat{L}_b\rho(t)\hat{L}_b^\dagger\right) = 1$ since we assume the completeness condition of $\hat{L}_a$ in the Eq. (5). In this case, the Eq. (8) is the same as the case without post-selection in the Eq. (6).

For the general $m < n$, the normalization factor $\sum_{b=1}^m \mathrm{Tr}[\hat{L}_b\rho(t + \delta t)\hat{L}_b^\dagger]$ is generally not one and depends on $\rho(t)$, and this is a non-linear differential equation of $\rho(t)$ different from the Lindblad master equation. The non-trivial normalization factor given by the post-selection process leads to a different differential equation of the density matrix.

# 4 The general properties of the generalized Lindblad equation

A dynamical equation of the density matrix should preserve the Hermitian, unit trace, and positive definiteness properties of the density matrix. In the following, we will show that these three properties of the density matrix are preserved under the evolution of our generalized Lindblad equation.

We find that only the second term on the left-hand side of the Eq. (8) is different from the original Lindblad master equation, and we know the original Lindblad master equation preserves these three properties. Hence, we can simply compare the difference between the second term on the right-hand side of our generalized Lindblad equation and that of the original Lindblad equation.

Firstly, since $\sum_{a=1}^m \frac{\hat{L}_a\rho(t)\hat{L}_a^\dagger}{\mathrm{Tr}\left(\sum_{b=1}^m \hat{L}_b\rho(t)\hat{L}_b^\dagger\right)}$ is Hermitian, and $\mathrm{Tr}\left[\sum_{a=1}^m \frac{\hat{L}_a\rho(t)\hat{L}_a^\dagger}{\mathrm{Tr}\left(\sum_{b=1}^m \hat{L}_b\rho(t)\hat{L}_b^\dagger\right)}\right] = \mathrm{Tr}\left[\frac{1}{2}\sum_{a=1}^n \{\hat{L}_a^\dagger\hat{L}_a, \rho(t)\}\right]$ $= 1$, it is easy to prove that $\rho(t)$ is Hermitian and preserves its unit trace along this evolution.

Secondly, the proof of positive definiteness is more involved, and it is as follows. We use the operator-sum representation of quantum channel [30]:

$$\rho(t + dt) = \epsilon_{dt}[\rho(t)] = \sum_{a=0}^n \hat{M}_a\rho(t)\hat{M}_a^\dagger. \tag{9}$$

Here,

$$\begin{cases} \hat{M}_0 &= \hat{I} + (-i\hat{H} + \hat{K})\,dt \\ \hat{M}_a &= \sqrt{\eta(t)}\hat{L}_a\sqrt{dt}, \quad a \neq 0 \end{cases} \tag{10}$$

with $\hat{L}_a$ representing quantum jump operators, and $\hat{I}$ representing the identity operator. Here, $\hat{K} = -\frac{1}{2}\eta(t)\sum_{a=1}^{n}\hat{L}_a^\dagger\hat{L}_a$. It is straightforward to see that this operator sum representation of quantum channel is equal to the Lindblad master equation the Eq. (6).

In our case, the density operator can be written in a similar operator-sum representation,

$$\rho(t+dt) = \epsilon_{dt}^{p}[\rho(t)] = \sum_{b=0}^{m}\hat{M}_b\rho(t)\hat{M}_b^\dagger \tag{11}$$

with

$$\begin{cases} \hat{M}_0 &= \hat{I} + (-i\hat{H} + \hat{K})\,dt \\ \hat{M}_b &= \sqrt{\alpha_t\eta(t)}\hat{L}_b\,\sqrt{dt}, \quad b = 1,...,m. \end{cases} \tag{12}$$

Here, $\hat{K} = -\frac{1}{2}\eta(t)\sum_{a=1}^{n}\hat{L}_a^\dagger\hat{L}_a$, and we also assume the completeness condition the Eq. (5). Here, the superscript $p$ in the Eq. (12) denotes the case with post-selection, and $\alpha_t = \left[\text{Tr}\left(\sum_{b=1}^{m}\hat{L}_b\rho(t)\hat{L}_b^\dagger\right)\right]^{-1}$ is the normalization factor attributed to post-selection. Also, since $m \le n$, the case where the system is being projected on the complete basis is included in the Eq. (11). We assume that at time $t$, the density matrix is positive definite, i.e. $\rho(t)$ can be decomposed as $\rho(t) = \sum_j p_j|\psi_j\rangle\langle\psi_j|$, $p_j \ge 0$. Thus, we see that

$$\langle\phi|\rho(t+dt)|\phi\rangle = \sum_{b=0}^{n}\langle\phi|\hat{M}_b\rho(t)\hat{M}_b^\dagger|\phi\rangle = \sum_{b=0}^{n}\sum_j p_j|\langle\psi_j|\hat{M}_b^\dagger|\phi\rangle|^2 \ge 0, \quad \forall\phi. \tag{13}$$

Therefore, we conclude that $\rho(t+dt)$ is positive definite provided that $\rho(t)$ is positive definite. This completes the proof.

## 5 Entanglement entropy in the MIFT

We then consider entropy dynamics in the MIPT. We here simply focus on studying the second Rényi entropy defined as: $S^{(2)} = -\log\left[\text{Tr}(\rho^2)\right]$. We first notice that the Eq. (8) is not able to describe entropy dynamics along this process. The reason is as follows. Since the measurement will produce different results, the density matrix of the system during the evolution process is the summation of the density matrix corresponding to the different measurement results, and the weight of each result is its probability of it. The reduced density matrix can be written as $\rho_A = \sum_c p_c\text{Tr}_{\bar{A}}\rho_c = \sum_c p_c\rho_{A,c}$. Here, $p_c$ is the probability of getting $\rho_c$, and it satisfies $\sum_c p_c = 1$. Therefore, it can be seen that the entanglement entropy comes from two parts: the entanglement entropy of the system corresponding to different measurement results $-\log\{\text{Tr}[\rho_{A,c}^2]\}$, and the entropy corresponding to the probability distribution of different measurement results $\{p_c\}$. In the MIPT, we are only concerned with the former part of entropy, and therefore we need to exclude the latter part. If the density matrix under the dynamic evolution described by the Eq. (8) is directly used in calculating entropy, we calculate the entanglement entropy of the mixed state obtained by the average of different measurement results:

$$S_A^{total} \equiv -\log\left\{\text{Tr}\left[\left(\sum_{c=1}^{m}p_c\rho_{A,c}\right)^2\right]\right\}. \tag{14}$$

Here, $p_c$ is the probality of getting $\rho_{A,c}$, and it satisfies $\sum_c p_c = 1$. Notice that not only the former part of entropy but also the latter part of entropy is included here. Therefore, the dynamics of entropy cannot be described only by the kinetic equation the Eq. (8).

To exclude the latter part of entropy in MIPT, we define a new type of entanglement entropy $S_A^{\text{new}}$ in this process as

$$S_A^{\text{new}} = -\log\left\{\sum_c \tilde{p}_c \left[\text{Tr}_A \rho_{A,c}^2\right]\right\}. \tag{15}$$

Here, $\tilde{p}_c = \frac{p_c^2}{\sum_{c'} p_{c'}^2}$, and it also satisfies $\sum_c \tilde{p}_c = 1$. To see that this definition of entanglement entropy the Eq. (15) does not take into account the entropy coming from the probability distribution $\{p_c\}$ in contrast to the Eq. (14), we consider a simple example. We assume that the system after evolution has probability $p_1 = \frac{1}{2}$ to be in $\rho_{A,1}$ with $\text{Tr}[\rho_{A,1}^2] = 1$, and probability $p_2 = \frac{1}{2}$ to be in $\rho_{A,2}$ with $\text{Tr}[\rho_{A,2}^2] = 1$. This means that in both cases, there is no entanglement entropy between subsystem $A$ and $B$. Thus, the entropy only comes from the probability distribution $\{p_c\} = \{\frac{1}{2}, \frac{1}{2}\}$. Also, we assume $\text{Tr}(\rho_{A,1}\rho_{A,2}) < 1$ which means $\rho_{A,1}$ and $\rho_{A,1}$ are not the same, then we obtain $S_A^{\text{new}} = -\log[\frac{1}{2} + \frac{1}{2}] = 0$ from the definition in the Eq. (15). However, if we use the $S_A^{\text{total}}$ defined in the Eq. (14) to calculate the entanglement entropy, we obtain $S_A^{\text{total}} = -\log[\frac{1}{4} + \frac{1}{2}\text{Tr}(\rho_{A,1}\rho_{A,2}) + \frac{1}{4}] = -\log[\frac{1}{2} + \frac{1}{2}\text{Tr}(\rho_{A,1}\rho_{A,2})] > 0$. Therefore, this definition of entanglement entropy includes the entropy coming from the classical distribution $\{p_c\}$. The proof of a more general case is written in the supplementary material [42].

Moreover, when all density matrix $\rho_{A,c}$ are mutually orthogonal, it is straightforward to prove that $S_A^{\text{total}} \geq S_A^{\text{new}}$ where the equality is taken when there is only one outcome with probability $p_1 = 1$. That means $S_A^{\text{total}} - S_A^{\text{new}}$ is non-negative as long as the entropy of probability distribution $\{p_c\}$ is non-zero. The details of this proof are written in the supplementary material [42].

# 6 The application of generalized Lindblad equation on the MIFT

In this section, we show that the entanglement entropy defined in the Eq. (15) can be obtained from a density matrix defined on a double space, and this double space density matrix's evolution is governed by a generalized Lindblad equation that we will propose later.

The dynamics of the second Rényi entropy can be mapped to the dynamics of a wave function defined on a double space, and therefore the second Rényi entropy dynamics is more straightforward when it is written on a double space [31, 32]. We denote the two copies of the system on double space as the left(L) and the right(R) system, and we use $\rho^D$ to denote the total density matrix of the double system. Given an initial density matrix $\rho = \sum_{mn} \rho_{mn}|m\rangle\langle n|$, the double state density matrix $\rho^D$ is given by:

$$\rho^D = \rho \otimes \rho = \sum_{mnst} \rho_{mn,st}^D |m\rangle_L \otimes |s\rangle_R \langle n|_L \otimes \langle t|_R \tag{16}$$

with $\rho_{mn,st}^D = \rho_{mn}\rho_{st}$. We can divide the system into subsystems $A$ and $B$, and derive the entanglement entropy of subsystem $A$ from $\rho_D$. Via a standard double space technique, the single system entropy $S_A \equiv -\log\left[\text{Tr}(\rho_A^2)\right]$ can be represented in the double space density matrix. Using $\text{Tr}_A(\rho_A^2) = \text{Tr}_{L_A,R_A}(X_A \rho_A \otimes \rho_A) = \text{Tr}_{L_A,R_A}\left[X_A \text{Tr}_{L_B,R_B}(\rho^D)\right]$, we have

$$S_A = -\log\{\text{Tr}_{L_A,R_A}\left[X_A \text{Tr}_{L_B,R_B}(\rho^D)\right]\}, \tag{17}$$

and it is illustrated in the Fig. (1).

Here, $\rho_A$ is the reduced density matrix of subsystem A calculated from the total density matrix as $\rho_A = \text{Tr}_B[\rho]$. Here, $\rho$ is the density matrix of the full system. $X$ is the swap operator defined as $X|\alpha\rangle_L|\beta\rangle_R = |\beta\rangle_L|\alpha\rangle_R$. Here, $X_A$ is the swap operator that acts on the subspace A. $L_A(L_B)$ is the $A(B)$ subsystem of the $L$ system, and $R_A(R_B)$ is the $A(B)$ subsystem of the $R$ system.

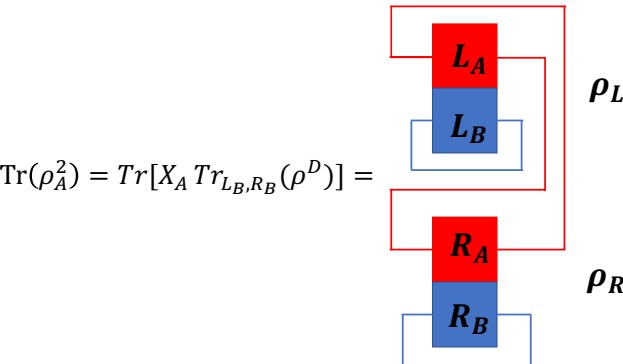

$$\mathrm{Tr}(\rho_A^2) = Tr[X_A \, Tr_{L_B,R_B}(\rho^D)] =$$

Figure 1: The schematic diagram of calculating entanglement entropy from the double state density matrix $\rho^D$.

When we consider the second Rényi entropy, to exclude the entropy coming from the classical distribution of different measurement results, we can enforce the condition that the L and the R system always collapse to the same state after measurement. In the double space, this condition means that the L and the R system are in the Einstein-Podolsky-Rosen (EPR) state. These above considerations motivate the equation of motion of the total density matrix $\rho^D$:

$$\frac{\partial \rho^D}{\partial t} = -i[\hat{H}^D, \rho^D(t)] + \eta(t) \sum_{a=1}^{n} \frac{\hat{L}_{a,L}\hat{L}_{a,R}\rho^D(t)\hat{L}_{a,L}^\dagger \hat{L}_{a,R}^\dagger}{\mathrm{Tr}\left(\sum_{b=1}^{n} \hat{L}_{b,L}\hat{L}_{b,R}\rho^D(t)\hat{L}_{b,L}^\dagger \hat{L}_{b,R}^\dagger\right)} - \frac{\eta(t)}{2} \sum_{a,b=1}^{n} \{\hat{L}_{a,L}^\dagger \hat{L}_{a,L}\hat{L}_{b,R}^\dagger \hat{L}_{b,R}, \rho^D(t)\}.$$

$$(18)$$

Here, $\hat{L}_{a,L}$ and $\hat{L}_{a,R}$ represent the jump operators acting on the $L$ and $R$ system respectively. Also, $\hat{H}^D = \hat{H}\otimes\hat{I} + \hat{I}\otimes\hat{H}$, and $\hat{I}$ denotes the identity operator. This equation can be obtained by replacing $\rho$ by $\rho^D$, $\hat{H}$ by $\hat{H}^D$ and $\hat{L}_a$ by $\hat{L}_{a,L}\hat{L}_{a,R}$ in the Eq. (8). Similar to the discussion in the previous section, the numerator of the second term on the right-hand side $\sum_{a=1}^{n} \hat{L}_{a,L}\hat{L}_{a,R}\rho^D(t)\hat{L}_{a,L}^\dagger \hat{L}_{a,R}^\dagger$ means that the double system is being projected on the EPR state since the quantum jump operator on the $L$ and $R$ system always being the same. It is a sharp contrast to the case where the numerator is chosen as $\sum_{a,b=1}^{n} \hat{L}_{a,L}\hat{L}_{b,R}\rho^D(t)\hat{L}_{a,L}^\dagger \hat{L}_{b,R}^\dagger$. Since $\sum_{a,b=1}^{n} \hat{L}_{a,L}\hat{L}_{b,R}\rho^D(t)\hat{L}_{a,L}^\dagger \hat{L}_{b,R}^\dagger$ describes the system been projected on the complete basis of the double space, whereas EPR states are partial basis of the double space. The denominator on the right hand side $\mathrm{Tr}\left(\sum_{b=1}^{n} \hat{L}_{b,L}\hat{L}_{b,R}\rho^D(t)\hat{L}_{b,L}^\dagger \hat{L}_{b,R}^\dagger\right)$ is the normalization factor resulting from post-selection. It is easy to see that this normalization factor is also non-trivial, making this equation a non-linear differential equation of $\rho^D$.

The generalized Lindblad equation Eq. (18) together with the entropy calculated through the double space technique the Eq. (17) are the central results of this paper. The essential property of the MIPT can be satisfied with this scheme, and the post-selection is explicitly embedded in the EPR state projection condition in the Eq. (18). These arguments rely on two steps:

1. Firstly, we will prove that the entropy formula the Eq. (17) for the evolved density matrix the Eq. (18) can be rewritten in the form of $S_A^{\mathrm{new}}$ defined in the Eq. (15). This argument fundamentally requires the EPR state projection structure achieved by double space and the choice of Lindblad jump operators in the Eq. (18).

2. Secondly, as we find previously, $S_A^{\mathrm{new}}$ in the Eq. (15) and the $S_A^{\mathrm{total}}$ in the Eq. (14) are conceptually different when describing the entanglement entropy. $S_A^{\mathrm{new}}$ can exclude the entropy coming from the classical distribution $\{p_c\}$ by excluding the cross term $p_c\rho_c p_{c'}\rho_{c'}$ with $c \neq c'$ when calculating the entropy. In the aforementioned literature about MIPT [8, 10, 15], such exclusions are implemented by post-selection, while here it was naturally embedded in the framework of the generalized Lindblad equation.

We will then give a proof of the first argument. Similar to the previous section, the evolution of the double system density matrix can be represented in the operator-sum form

$$\rho^D(t + \mathrm{d}t) = \epsilon_{\mathrm{d}t}[\rho^D(t)] = \sum_{b=0}^{m} \hat{M}_b^D \rho_0^D(t) \hat{M}_b^{D\dagger} \tag{19}$$

with

$$\begin{cases} \hat{M}_0^D & = \hat{I}^D + (-i\hat{H}^D + \hat{K}^D)\, \mathrm{d}t \\ \hat{M}_b^D & = \sqrt{\alpha_t^D \eta(t)} \hat{L}_{b,L} \hat{L}_{b,R}\, \sqrt{\mathrm{d}t}, \quad b = 1, ..., m. \end{cases} \tag{20}$$

Here, $\hat{K}^D = \hat{K}_L + \hat{K}_R = -\frac{1}{4}\eta(t) \sum_{a=1}^{n} \hat{L}_{a,L}^{\dagger} \hat{L}_{a,L} - \frac{1}{4}\eta(t) \sum_{a=1}^{n} \hat{L}_{a,R}^{\dagger} \hat{L}_{a,R}$, and we also assume the completeness condition the Eq. (5). Then, by only keeping the first order of d$t$, we find that $\hat{M}_b^D$ can be rewritten as

$$\begin{cases} \hat{M}_0^D & = \hat{M}_{0,L} \otimes \hat{M}_{0,R} \\ \hat{M}_b^D & = \hat{M}_{b,L} \otimes \hat{M}_{b,R}, \quad b = 1, ..., m \end{cases} \tag{21}$$

with

$$\hat{M}_0 = \hat{I} + (-i\hat{H} + \hat{K})\, \mathrm{d}t$$

$$\hat{M}_b = \left[\alpha_t^D \eta(t)\right]^{\frac{1}{4}} \hat{L}_b (\mathrm{d}t)^{\frac{1}{4}},$$

and $\alpha_t^D = \left[\mathrm{Tr}\left(\sum_{b=1}^{n} \hat{L}_{b,L} \hat{L}_{b,R} \rho^D(t) \hat{L}_{b,L}^{\dagger} \hat{L}_{b,R}^{\dagger}\right)\right]^{-1}$ is the normalization factor attributed to post-selection in the double space. Then the entanglement entropy calculated from $\rho_D$ can be written as

$$\begin{aligned} S_A^D &= -\log\left\{ \mathrm{Tr}_{L_A,R_A}\left[X_A \mathrm{Tr}_{L_B,R_B}(\sum_{b=0}^{m} \hat{M}_b^D \rho_0^D \hat{M}_b^{D\dagger})\right]\right\} \\ &= -\log\{\sum_{b=0}^{m} \mathrm{Tr}_{L_A,R_A}[X_A \mathrm{Tr}_{L_B,R_B}(\hat{M}_{b,L}\rho_0\hat{M}_{b,L}^{\dagger}) \otimes \quad (\hat{M}_{b,R}\rho_0\hat{M}_{b,R}^{\dagger})]\} \\ &= -\log\left\{\sum_{b=0}^{m} \tilde{p}_b \mathrm{Tr}_{L_A,R_A}\left[X_A \rho_{A,b} \otimes \rho_{A,b}\right]\right\} \\ &= -\log\left\{\sum_{b=0}^{m} \tilde{p}_b \mathrm{Tr}_A\left[\rho_{A,b}^2\right]\right\} \end{aligned} \tag{22}$$

with $\tilde{p}_b = \left[\mathrm{Tr}\left(\hat{M}_b\rho_0(t)\hat{M}_b^{\dagger}\right)\right]^2$ and $\rho_{A,b} = \frac{\mathrm{Tr}_B\left[\hat{M}_b\rho_0(t)\hat{M}_b^{\dagger}\right]}{\mathrm{Tr}\left(\hat{M}_b\rho_0(t)\hat{M}_b^{\dagger}\right)}$. Also, if we define $\mathrm{Tr}\left(\hat{L}_b\rho_0(t)\hat{L}_b^{\dagger}\right) = p_b$, then we have $\tilde{p}_b = \alpha_t^D \eta(t)\left[\mathrm{Tr}\left(\hat{L}_b\rho_0(t)\hat{L}_b^{\dagger}\right)\right]^2 = \eta(t)\frac{p_b^2}{\sum_c p_c^2}$ for $b = 1, ..., n$. When we consider the case $\eta(t) = 1$,which means that the probability of the system being measured per unit time is 1, this definition of $\tilde{p}_b$ is consistent with that of the new type of entanglement entropy $S_A^{\mathrm{new}}$ in the Eq. (15). This completes the proof.

# 7   Numerical results

In this section, we numerically study the second Rényi entropy dynamics of a 1D hard-core Bose Hubbard [33] system to demonstrate that there is an entanglement phase transition under the Eq. (18) and (17). The Hamiltonian of the hard-core Bose Hubbard system is

$$\hat{H} = -J \sum_{\langle i,j \rangle} \hat{b}_i^{\dagger} \hat{b}_j + U \sum_{\langle i,j \rangle} \hat{n}_i \hat{n}_j. \tag{23}$$

Here, $J$ is the strength of the nearest neighbor hopping, and $U$ is the strength of the nearest neighbor interaction.

The system is driven by the generalized Lindblad equation the Eq. (18) in the double system, and we set $\eta(t) = \gamma$ as a time-independent measurement rate. Also, we set the projection measurements as

$$\hat{L}_{i,0} = \frac{1}{\sqrt{L}}(1 - \hat{n}_i), \ \hat{L}_{i,1} = \frac{1}{\sqrt{L}}\hat{n}_i. \tag{24}$$

Here $i = 1, 2, ..., N_s$, and $N_s$ is the total number of sites. Also, we further normalize the projection operators to satisfy the completeness condition of the measurement basis in the Eq. (5). These projection operators mean that the environment is measuring the particle number on each site by projecting it on one of the particle number basis ($|0\rangle, |1\rangle$). Here, $|1\rangle$ denotes the site is occupied, and $|0\rangle$ denotes the site is unoccupied.

In our following numerical calculation, we set $J = U = 1, N_s = 6, N_b = 3$. $N_b$ is the total number of the hard-core boson. We denote the left half of the system as subsystem $A$ and the rest of it as subsystem $B$. We then calculate the entanglement entropy $S_A$ defined in the Eq. (17). We choose the initial state as a product state in the particle number basis $|000111\rangle$.

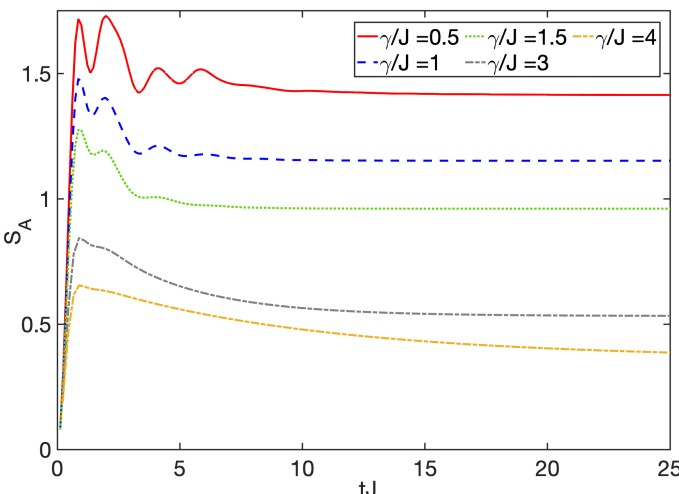

Figure 2: The dynamics of the entanglement entropy $S_A$ as a function of $tJ$. $\gamma$ is the measurement rate. Different curves have different $\gamma$ in the unit of $J$. Here, $U = J$ and the number of sites $N_s = 6$, and the number of bosons $N_b = 3$, and subsystem size $L_A = 3$.

As shown in Fig. 2, the entanglement entropy $S_A$ between subsystems $A$ and $B$ first quickly increases as expected in a normal chaotic system. However, it then decreases and saturates to a non-zero value. It indicates that measurements and the following post-selection process decrease the entanglement between subsystems. Also, we prove in the supplementary material [42] that the entanglement entropy is the same regardless of whether one computes partial trace over the subsystem $A$ or subsystem $B$, and this explains why entanglement entropy $S_A$ is symmetric with respect to the half system size $L_A$ axis ($L_A = N_s/2 = 3$) in Fig. 2.

Moreover, from the result in Fig. 3, we find that when the measurement rate is small ($\gamma/J = 0.5$) and $L_A < N_s/2$, the entanglement entropy between the two subsystems A and B is almost linear in system size $L_A$. Whereas when the measurement rate is large ($\gamma/J = 5$) and $L_A < N_s/2$, the entanglement entropy is almost flat as $L_A$ changes. In the MIPT, as the measurement rate $\gamma$ increases, the entanglement entropy between the two subsystems $A$ and $B$ will change from volume-law to area-law, and our results are consistent with it. Thus, our result is in line with the understanding that there is an entanglement phase transition in this process. Therefore, we use a concrete example to show that our generalized Lindblad equation the Eq. (18) can describe

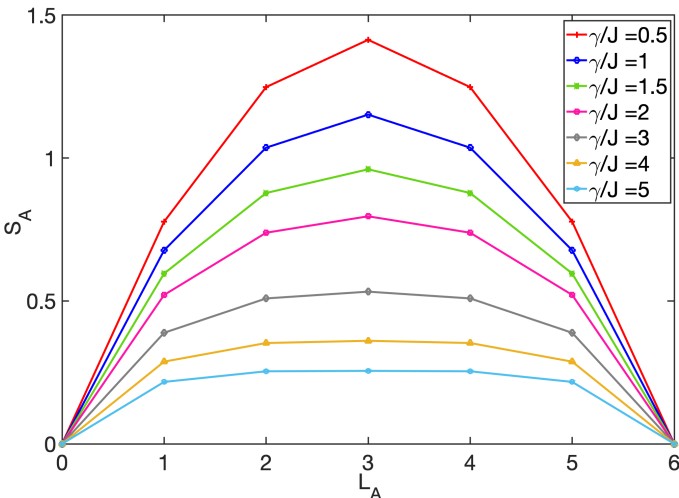

Figure 3: Saturation value of entanglement entropy $S_A$ of the subsystem $A$ with different subsystem size $L_A$. Different curves have different $\gamma$ in the unit of $J$. Here, $N_s = 6$, $N_b = 3$, $U/J = 1$.

an entanglement phase transition as the measurement rate increases. The detailed information about the numerical integration technique used in producing Fig. 2 and Fig. 3 is added in the supplementary material [42].

Also, the numerical results of entanglement entropy calculated from the single-copy master equation the Eq. (8) are added in the supplementary material [42] for comparison. We find that there is no entanglement phase transition in this process, and there is no surprise since we mentioned in the section 5 that the entanglement entropy calculated from the single-copy master equation is $S_A^{\text{total}}$ defined in the Eq. (14). Since $S_A^{\text{total}}$ also includes the entropy corresponding to the probability distribution of different measurement results, there is no entanglement phase transition regarding this entanglement entropy.

# 8 Discussions

In this paper, we derive a generalized Lindblad equation for describing the dynamics of the system under the measurement process followed by post-selection. We emphasize the post-selection is essential in the non-linear differential the Eq. (18). Also, the generalized Lindblad equation the Eq. (18) preserves the Hermitian, unit trace, and positive definiteness of the density matrix. Furthermore, we generalize it to describe the second Rényi entropy dynamics in the MIPT, and we use a concrete model to demonstrate that our equation can indeed describe this entanglement transition as the measurement rate increases. Also, our generalized Lindblad equation can be simulated by the quantum trajectory methods [35–41] as the original Lindblad equation, and we consign the details to the Supplemental Material [42]. Therefore, the numerical simulation of these dissipative dynamics is feasible.

Moreover, the post-selection provides information about the probability distribution of measurement results. By using the information of the measurement results, we project the system to some pure states. Hence, we decrease the entropy of the system. Moreover, the Holevo information [34] defined as $\chi := S_v(\rho) - \sum_i p_i S_v(\rho_i)$ actually measures how much entropy on average is reduced once we learn the the distribution $\{p_i\}$. Here, $S_v$ is the von Neumann entropy. Thus, if we measure von Neumann entropy in the MIPT, we find that the amount of entropy decreased by measurement and post-selection is just the Holevo information. The decrease of entropy by the measurement and post-selection process is owing to gaining accessible information about the

measurement results.

The results we have presented here suggest some further directions that are worth exploring. It will be enlightening to analytically calculate the entropy from our generalized Lindblad equation. Also, It will be interesting to explore the Holevo information in the MIPT and try to understand this phase transition from the perspective of getting accessible classical information. It is also interesting to experimentally realize this generalized Lindblad equation by coupling the system to a bath and designing the form of interaction to satisfy the normalization factor of our generalized Lindblad equation. Therefore, we may further find some direct experimental access to the entanglement phase transition.

# Acknowledgments

We thank Hui Zhai, Pengfei Zhang, and Tian-Gang Zhou for helpful discussion and for carefully reading the manuscript.

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

# A  The definition of the double space density matrix from an initial single space density matrix

Given an initial density matrix $\rho = \sum_{mn} \rho_{mn}|m\rangle\langle n|$, the double space density matrix $\rho^D$ is given by:

$$\rho^D = \sum_{mnst} \rho^D_{mn,st}|m\rangle_L \otimes |s\rangle_R {}_L\langle n| \otimes_R \langle t| \tag{25}$$

with $\rho^D_{mn,st} = \rho_{mn}\rho_{st}$. Here, we assume that the initial double space density matrix is the direct product of two same single space density matrices. $\rho^D$ is illustrated in the Fig. (4).

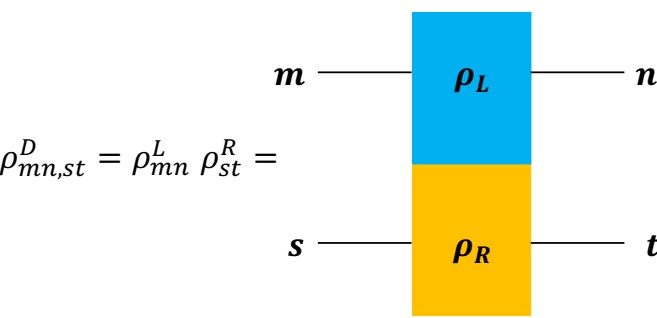

Figure 4: The schematic diagram of the definition of $\rho^D$.

# B  Three different definitions of entanglement entropy in the measurement-induced phase transition

In the evolution process together with measurements, measurements will produce different results. Thus, the density matrix in this process is the summation of the density matrix corresponding to the different measurement results, and each weight of the sum is the probability of that result. Therefore, we can write the density matrix as $\rho = \sum_c p_c \rho_c$. Here, $c$ represents the different cases of evolution due to different measurement results. If we calculate the entanglement entropy directly from this mixed state density matrix, we obtain

$$S^{\text{total}} = -\log\left\{\text{Tr}\left[\left(\sum_c p_c\rho_c\right)^2\right]\right\}. \tag{26}$$

Also, we propose a new definition of the second-order Rényi entropy in our paper:

$$S^{\text{new}} = -\log\left\{\sum_c \tilde{p}_c \text{Tr}[\rho_c^2]\right\}. \tag{27}$$

Here, $\tilde{p}_c = \frac{p_c^2}{\sum_{c'} p_{c'}^2}$. There is also another kind of the second-order Rényi entropy that people have used in the MIPT [8, 10]:

$$S^{\text{old}} = -\sum_c p_c \log\{\text{Tr}[\rho_c^2]\}. \tag{28}$$

## C   The comparison of three different definitions of entanglement entropy in the measurement-induced phase transition

The entropy in MIPT comes from two parts: the entropy of the system corresponding to different measurement results $-\log\{\text{Tr}[\rho_c^2]\}$, and the entropy of the probability distribution of different measurement results $\{p_c\}$. In the MIPT, we only care about the former part of entropy, and therefore we need to exclude the latter part.

We will show that both the $S^{\text{new}}$ and $S^{\text{old}}$ can exclude the latter part of entropy, whereas $S^{\text{total}}$ can not. For instance, we assume that the system after evolution has a probability $p_c$ to be in the case $c$, and the density matrix in the case $c$ is $\rho_c$. Here, $\forall c, p_c > 0$, and $\sum_{c=1}^{n} p_c = 1$ with $n > 1$. We assume that every $\rho_c$ satisfies $\text{Tr}[\rho_c^2] = 1$, and this means that in each case, $\rho_c$ is a pure state density matrix. Meanwhile, we assume $\text{Tr}(\rho_c \rho_{c'}) < 1$ for $c \neq c'$, and this means $\rho_c$ and $\rho_{c'}$ are not the same. Thus, the entropy only comes from the probability distribution $\{p_c\} = \{p_1, p_2, ..., p_n\}$. From the definition in the Eq. (26), we have

$$
\begin{aligned}
S^{\text{total}} &= -\log\{\text{Tr}[(\sum_{c=1}^{n} p_c \rho_c)^2]\} \\
&= -\log\left[\sum_{c=1}^{n} p_c^2 \text{Tr}(\rho_c^2) + 2\sum_{c<c'} p_c p_{c'} \text{Tr}(\rho_c \rho_{c'})\right] \\
&> -\log\left[\sum_{c=1}^{n} p_c^2 \text{Tr}(\rho_c^2) + 2\sum_{c<c'} p_c p_{c'}\right] \\
&= -\log\left[\sum_{c=1}^{n} p_c^2 + 2\sum_{c<c'} p_c p_{c'}\right] \\
&= -\log\left[(\sum_{c=1}^{n} p_c)^2\right] = 0.
\end{aligned}
\tag{29}
$$

Here, we use $\text{Tr}(\rho_{A,c} \rho_{A,c'}) < 1$ for $c \neq c'$ to get the first inequality. Thus, we obtain $S^{\text{total}} > 0$.

In comparison with $S^{\text{total}}$, from the definition in the Eq. (27), we have $S^{\text{new}} = -\log[\sum_i \tilde{p}_i \text{Tr}(1)] = 0$. Also, from the definition in the Eq. (28), we have $S^{\text{old}} = -\sum_i p_i \log[\text{Tr}(1)] = 0$. Therefore, we find that $S^{\text{total}}$ takes into account the entropy of probability distribution $\{p_c\} = \{p_1, p_2, ..., p_n\}$, whereas $S^{\text{new}}$ and $S^{\text{old}}$ do not.

Also, we find that if the density matrix in each case has the same purity and occurs with the same probability, then $S^{\text{new}}$ and $S^{\text{old}}$ are the same. In this case, we have $p_c = \frac{1}{n}, \text{Tr}[\rho_c^2] = a$ for $c = 1, ..., n$. Then, we have $S^{\text{new}} = -\log(\sum_{c=1}^{n} \frac{1}{n} a) = -\log(a)$, and $S^{\text{old}} = -\sum_{c=1}^{n} \frac{1}{n} \log a = -\log[a^{(n \times \frac{1}{n})}] = -\log(a)$. Thus, we find that $S^{\text{new}}$ and $S^{\text{old}}$ are the same in this case.

## D   The proof of an inequality between $S^{\text{total}}$ and $S^{\text{new}}$

In this section, we will prove that when all density matrix $\rho_c$ are mutually orthogonal, we have $S^{\text{total}} \geq S^{\text{new}}$ where the equality is taken when there is only one outcome with probability $p_1 = 1$.

Start from the definition $S^{\text{total}}$, we have

$$
\begin{aligned}
S^{\text{total}} &= -\log\{\text{Tr}[(\sum_{c=1}^{n} p_c \rho_c)^2]\} \\
&= -\log\left[\sum_{c,c'=1}^{n} p_c p_{c'} \text{Tr}(\rho_c \rho_{c'})\right] \\
&= -\log\left[\sum_{c}^{n} p_c^2 \text{Tr}(\rho_c^2)\right] \\
&\geq -\log\left[\sum_{c}^{n} \tilde{p}_c \text{Tr}(\rho_c^2)\right] \\
&= S^{\text{new}}.
\end{aligned}
\tag{30}
$$

Here, we use the $\text{Tr}(\rho_c \rho_{c'}) = 0$ for $c \neq c'$ to obtain the third equality. Here, Also, we use $\tilde{p}_c = \frac{p_c^2}{\sum_{c'} p_{c'}^2} \geq p_c^2$ to obtain the fourth line. The equality in the fourth line is taken when there is only one outcome with probability $p_1 = 1$. Thus, we complete the proof.

## E  The quantum trajectories method

In this section, we introduce the quantum trajectories method regarding our generalized Lindblad equation the Eq. (18). The quantum trajectories method involves rewriting the master equation as a stochastic average over individual trajectories. It is an efficient tool for numerically simulating dissipative dynamics. Similar to the original Lindblad Master equation, our generalized Lindblad equation can be expressed in an alternative form:

$$
\frac{\partial \rho^D}{\partial t} = -i\left(\hat{H}_{\text{eff}}^D \rho(t) - \rho(t)\hat{H}_{\text{eff}}^{D\dagger}\right) + \eta(t)\sum_{a=1}^{n} \frac{\hat{L}_{a,L}\hat{L}_{a,R}\rho^D(t)\hat{L}_{a,L}^\dagger \hat{L}_{a,R}^\dagger}{\text{Tr}\left(\sum_{b=1}^{n} \hat{L}_{b,L}\hat{L}_{b,R}\rho^D(t)\hat{L}_{b,L}^\dagger \hat{L}_{b,R}^\dagger\right)}.
\tag{31}
$$

Here $\hat{H}_{\text{eff}}^D$ is the effective Hamiltonian defined as

$$
\hat{H}_{\text{eff}}^D = \hat{H} \otimes \hat{I} + \hat{I} \otimes \hat{H} - i\frac{\eta(t)}{2}\sum_{a,b} \hat{L}_{a,L}^\dagger \hat{L}_{a,L}\hat{L}_{b,R}^\dagger \hat{L}_{b,R}.
\tag{32}
$$

First, we start from a double system initial state $|\phi^D(t)\rangle$, and compute its evolution under the effective Hamiltonian after a small time step $\delta t$:

$$
|\phi^{(1),D}(t+\delta t)\rangle = (1 - iH_{\text{eff}}^D \delta t)|\phi^D(t)\rangle.
\tag{33}
$$

Then, we compute the norm of this wave function at time $t + \delta t$.

$$
\langle \phi^{(1),D}(t+\delta t)|\phi^{(1),D}(t+\delta t)\rangle \equiv 1 - \delta p
\tag{34}
$$

where

$$
\delta p = \eta(t)\langle \phi^D(t)|i(H_{\text{eff}}^D - H_{\text{eff}}^{D\dagger})|\phi^D(t)\rangle \delta t = \eta(t)\langle \phi^D(t)|\sum_{a,b} \hat{L}_{a,L}^\dagger \hat{L}_{a,L}\hat{L}_{b,R}^\dagger \hat{L}_{b,R}|\phi^D(t)\rangle \delta t.
\tag{35}
$$

Here, we assume the completeness condition of the Eq. (5). Thus, we further have

$$
\delta p = \eta(t)\delta t.
\tag{36}
$$

Second, we choose the propagated state stochastically in the following manner:
1. With probability $1 - \delta p$, the wave function at $t + \delta t$ is chosen as the one that evolves under the effective Hamiltonian with a normalization factor accordingly:

$$|\phi^D(t + \delta t)\rangle = \frac{|\phi^{(1),D}(t + \delta t)\rangle}{\sqrt{1 - \delta p}}. \tag{37}$$

2. With probability $\delta p$, the wave function at $t + \delta t$ is chosen as the one that jumps to some particular quantum channel $a$:

$$|\phi^D(t + \delta t)\rangle = \frac{\hat{L}_{a,L}\hat{L}_{a,R}|\phi^D(t)\rangle}{\sqrt{\frac{\delta p_a}{\delta t}}} \tag{38}$$

where $\delta p_a = \delta t \langle \phi^D(t)|\hat{L}_{a,R}^\dagger \hat{L}_{a,L}^\dagger \hat{L}_{a,L}\hat{L}_{a,R}|\phi^D(t)\rangle$. Here, each quantum channel $a$ is chosen with a probability $\Pi_a = \frac{\delta p_a}{\sum_b \delta p_b}\delta p$.

Since from the prescription above, the propagation of the initial density matrix $\rho^D(t) = |\phi^D(t)\rangle\langle\phi^D(t)|$ in a given time step is:

$$\overline{\rho^D(t + \delta t)} = (1 - \delta p)\frac{|\phi^{(1),D}(t + \delta t)\rangle}{\sqrt{1 - \delta p}}\frac{\langle\phi^{(1),D}(t + \delta t)|}{\sqrt{1 - \delta p}} + \delta p \sum_a \Pi_a \frac{\hat{L}_{a,L}\hat{L}_{a,R}|\phi^D(t)\rangle}{\sqrt{\frac{\delta p_a}{\delta t}}}\frac{\langle\phi^D(t)|\hat{L}_{a,R}^\dagger \hat{L}_{a,L}^\dagger}{\sqrt{\frac{\delta p_a}{\delta t}}}$$

$$= |\phi^{(1),D}(t + \delta t)\rangle\langle\phi^{(1),D}(t + \delta t)| + \eta(t)\delta t \sum_a \frac{\hat{L}_{a,L}\hat{L}_{a,R}|\phi^D(t)\rangle\langle\phi^D(t)|\hat{L}_{a,R}^\dagger \hat{L}_{a,L}^\dagger}{\text{Tr}\left(\sum_{b=1}^n \hat{L}_{b,L}\hat{L}_{b,R}\rho^D(t)\hat{L}_{b,L}^\dagger \hat{L}_{b,R}^\dagger\right)}$$

$$= -i\left(\hat{H}_{\text{eff}}^D\rho(t) - \rho(t)\hat{H}_{\text{eff}}^{D\dagger}\right)\delta t + \eta(t)\delta t \sum_a \frac{\hat{L}_{a,L}\hat{L}_{a,R}|\phi^D(t)\rangle\langle\phi^D(t)|\hat{L}_{a,R}^\dagger \hat{L}_{a,L}^\dagger}{\text{Tr}\left(\sum_{b=1}^n \hat{L}_{b,L}\hat{L}_{b,R}\rho^D(t)\hat{L}_{b,L}^\dagger \hat{L}_{b,R}^\dagger\right)}. \tag{39}$$

Here, the $\overline{\rho^D(t + \delta t)}$ denotes a statistical average over trajectories. It is straightforward to see that the stochastic propagation given by this quantum trajectories method is equivalent to our generalized Lindblad equation Eq. (18) after taking a stochastic average over trajectories.

# F  The entanglement entropy of complement subsystems

In this section, we will prove that the entanglement entropy calculated from the entropy formula the Eq. (17) for the evolved density matrix the Eq. (18) is the same regardless of whether one computes partial trace over the subsystem $A$ or subsystem $B$. Here, subsystems $A$ and $B$ are complement subsystems of the total system. From the Eq. (22), we have

$$S_A^D = -\log\left\{\sum_{b=0}^m \tilde{p}_b \text{Tr}_A\left[\rho_{A,b}^2\right]\right\}. \tag{40}$$

Also, if we change the subsystem $A$ to its complement subsystem B, we have

$$S_B^D = -\log\left\{\sum_{b=0}^m \tilde{p}_b \text{Tr}_B\left[\rho_{B,b}^2\right]\right\}. \tag{41}$$

with $\tilde{p}_b = \left[\text{Tr}\left(\hat{M}_b\rho_0(t)\hat{M}_b^\dagger\right)\right]^2$ and $\rho_b = \frac{\hat{M}_b\rho_0(t)\hat{M}_b^\dagger}{\text{Tr}\left(\hat{M}_b\rho_0(t)\hat{M}_b^\dagger\right)}$. Here, each $\rho_b$ is pure if we start from a pure state. For a pure state density matrix, using the Schmidt decomposition, we can prove that

$\mathrm{Tr}_A\left[\rho_{A,b}^2\right] = \mathrm{Tr}_B\left[\rho_{B,b}^2\right]$, thus we obtain

$$
\begin{aligned}
S_B^D &= -\log\left\{\sum_{b=0}^{m}\tilde{p}_b\mathrm{Tr}_B\left[\rho_{B,b}^2\right]\right\} \\
&= -\log\left\{\sum_{b=0}^{m}\tilde{p}_b\mathrm{Tr}_A\left[\rho_{A,b}^2\right]\right\} \\
&= S_A^D.
\end{aligned}
\tag{42}
$$

# G  Numerical results of the entropy computed from the single-copy master equation

In this section, we will give some numerical results about the system driven by the generalized Lindblad equation the Eq. (8) in the original single system, and we set $\eta(t) = \gamma$ as a time-independent measurement rate. Also, we set the projection measurements as

$$
\hat{L}_{i,0} = \frac{1}{\sqrt{L}}(1-\hat{n}_i),\ \hat{L}_{i,1} = \frac{1}{\sqrt{L}}\hat{n}_i.
\tag{43}
$$

The Hamiltonian of is also the hard-core Bose Hubbard system

$$
\hat{H} = -J\sum_{\langle i,j\rangle}\hat{b}_i^\dagger\hat{b}_j + U\sum_{\langle i,j\rangle}\hat{n}_i\hat{n}_j.
\tag{44}
$$

as the section 7 for comparison. Here, $J$ is the strength of the nearest neighbor hopping, and $U$ is the strength of the nearest neighbor interaction.

In our following numerical calculation, we set $J = U = 1, N_s = 6, N_b = 3$. $N_b$ is the total number of the hard-core boson. We denote the left half of the system as subsystem $A$ and the rest of it as subsystem $B$. We then calculate the entanglement entropy $S_A$ defined in the Eq. (17). We choose the initial state as the a product state in the particle number basis $|000111\rangle$.

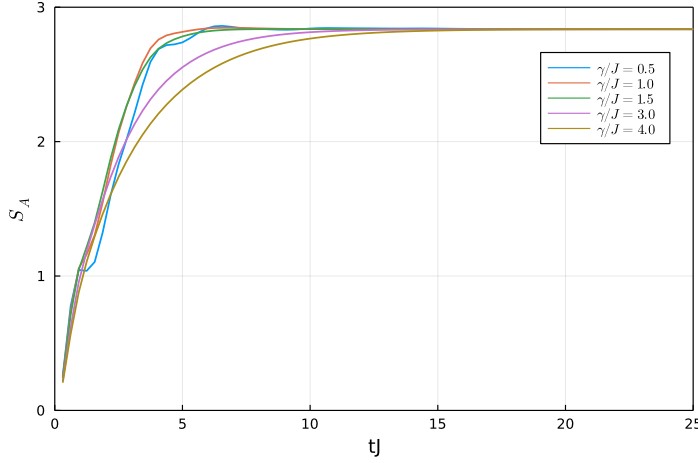

Figure 5: The dynamics of the entanglement entropy $S_A$ as a function of $tJ$. $\gamma$ is the measurement rate. Different curves have different $\gamma$ in the unit of $J$. Here, $U = J$ and the number of sites $N_s = 6$, and the number of bosons $N_b = 3$, and subsystem size $L_A = 3$. The density matrix is evolved by the single-copy master equation the Eq. (8).

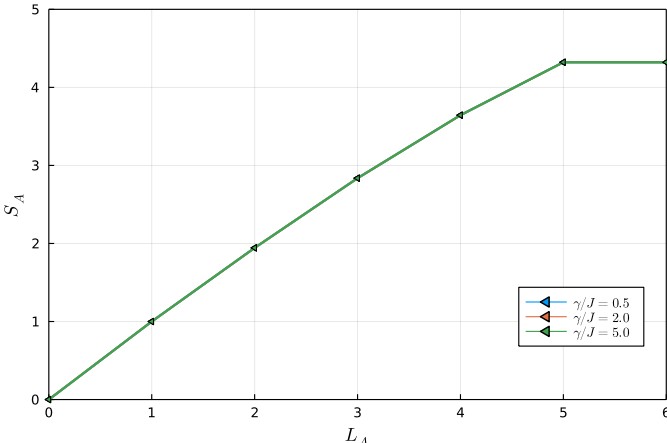

Figure 6: Saturation value of entanglement entropy $S_A$ of the subsystem $A$ with different subsystem size $L_A$. Different curves have different $\gamma$ in the unit of $J$. Here, $N_s = 6$, $N_b = 3$, $U/J = 1$. The density matrix is evolved by the single-copy master equation the Eq. (8).

As shown in Fig. 5, the entanglement entropy $S_A$ between subsystems $A$ and $B$ increases as expected in a normal chaotic system, and it then saturates to a non-zero value. However, compared with the dynamics of the entanglement entropy $S_A$ in the Fig. 2, it does not decrease in this process, and the saturation value of $S_A$ shown in the Fig. 6 is the same as measurement rate changes.

Therefore, we find that there is no entanglement phase transition in this process, and there is no surprise since we have mentioned in the section 5 that the entanglement entropy calculated from the single-copy master equation is $S_A^{\text{total}}$ defined in the Eq. (14) as

$$S_A^{\text{total}} \equiv -\log\left\{ \text{Tr}\left[ \left( \sum_{c=1}^{m} p_c \rho_{A,c} \right)^2 \right] \right\}. \tag{45}$$

Here, $p_c$ is the probability of getting $\rho_{A,c}$, and it satisfies $\sum_c p_c = 1$. Since $S_A^{\text{total}}$ also includes the entropy corresponding to the probability distribution of different measurement results, there is no entanglement phase transition regarding this entanglement entropy.

# H   The detail information about the numerical integration technique used in producing Fig. 2 and Fig. 3

We used the Runge-Kutta 4th-order (RK4) method for the approximate solutions of simultaneous nonlinear equations about the double system density matrix. For the Fig. 2 and Fig. 3, we used $N_t = 11000$ time steps.