# Peer review of "Generalized Lindblad Master Equation for Measurement-Induced Phase Transition"

_SciPost Physics Core_

## Round 1 · Referee Report · Anonymous (Referee 1) · 2022-12-9

Strengths

1) The paper is interesting and scientifically sound 2) The topic is extremely timely 3) Needed calculations are carried out explicitly

Weaknesses

  • Initial sections are just a re-derivation of well known results and in doing so some misleading error have been made (see report)
  • not very clear connections of the results with MIPT physics

Report

In this paper the author propose a generalized Lindblad equation to compute the seconde Rényi entropy in systems subject to the combined action of unitary evolution and measurement.

The paper is interesting, scientifically sound and the topic is extremely timely. However I feel that some modifications needs to be done before the publication. My main concern is about Sec.2 where some well known results are re-derived and some confusion is made in Sec.4 . I formulate here below the points that need to be addressed :

1) The Lindblad ME (6) arises naturally when considering a system being subject to both unitary dynamics and measurement. Indeed it is well known that the average dynamics of this kind of systems gives rise to a Lindblad ME both in case of a projective measurement and weak measurement. The derivation that the author propose (from Eq.1 to Eq.6) requires that $\sum_{\alpha=1}^n L^\dagger_\alpha L_\alpha = \mathbb{I}$ . This fact imply that the anti commutator part of the Lindblad ME is completely trivial and amounts to an identity matrix.

However, when using an operator-sum representation [Eq.(9,10)] to derive the Lindblad ME one can notice that the equality $\sum_{\alpha=1}^n L^\dagger_\alpha L_\alpha = \mathbb{I}$ does NOT need to be satisfied in general [in Eq.(10) one can simply keep $M_0$ and $M_1$ as done in the references I listed below]: the operator-sum representation is still well defined since $\sum_{\alpha=0}^N M_\alpha^\dagger M_\alpha = \mathbb{I}$ and the result is a Lindblad ME with a non-trivial anticommutator (giving rise to the so called non-Hermitian effective Hamiltonian). This is the typical situation arising from a weak measurement approach (coupling the system to an ancilla and then performing a projective measurement on it). I thus suggest the author to clarify the difference between Eq.(6) [that strongly rely on the assumption $\sum_{\alpha=1}^n L^\dagger_\alpha L_\alpha = \mathbb{I}$ and the “weak measurement” scenario where$\sum_{\alpha=1}^n L^\dagger_\alpha L_\alpha \neq \mathbb{I}$. Maybe the author can find useful the paradigmatic cases studied in the list of references.

[1] https://quantum-journal.org/papers/q-2021-08-19-528/ [2] https://journals.aps.org/prb/abstract/10.1103/PhysRevB.103.224210 [3] https://journals.aps.org/prresearch/abstract/10.1103/PhysRevResearch.2.033512

2) I do not understand the connection of the numerical results with the MIPT. What I mean is that in order to claim that a system undergoes (or not) a MIPT one needs to study how the entanglement entropy scales as the size of the system is increased. Typically this requires to simulate large L and to perform a scaling analysis. I suggest the author to reshape Sec.7 and be more prudent about how the results are supportive or not of a MIPT. What we can say is that, as expected, we observe a reduction of the entropy as \gamma/J is increased.

3) As a sanity check of the generalized Lindblad equation I suggest the author to compare the results in Fig.2 with a standard quantum trajectory evolution performed on a Lindblad ME (defined in the standard physical Hilbert space and not in the doubled space) with jump operators as in Eq.(24) and Hamiltonian as in Eq.(23). Then, at each time the entropy can be computed for each trajectory and the average should correspond to their results. This is a cheap simulation that would clarify the results. This “direct” method would directly give the correct contribution of the entropy as in Eq.(15).

4) Some typos need to be fixed as, e.g., MIFT is written instead of MIPT in different points in the manuscript

Requested changes

1) reorganize the initial Sections of the manuscript 2) explain better the results and their connections with MIPT 3) check the correctness of the results with a "direct" quantm trajectory approach 4) fix the typos

  • validity: good
  • significance: good
  • originality: good
  • clarity: good
  • formatting: good
  • grammar: good

Author:  Yi-Neng Zhou  on 2023-01-03  [id 3201]

(in reply to Report 1 on 2022-12-09)
Category:
remark
correction

  1. I have added the discussion in section 3 of the main text and Sec.I of the supplementary about the difference between Eq. (6) and the “weak measurement” scenario which generally does not assume the projection on a complete basis. And I have cited the list of references that the referee has recommended as the ref [12],[31],[32]. In deriving the Lindblad-like equation in the section.2, we assume that the measurement is performed on a complete basis, thus the anti-commutator part of the Lindblad master equation is trivial and amounts to an identity matrix, Thus, Eq. (6) is a special case of the general Lindblad equation. Also, the probabilistic measurement process in Eq. (4) leads to the anti-commutator part of the Lindblad master equation amounts to an identity matrix. If we do not assume this probabilistic measurement process, the system under unitary evolution together with the measurement process can be described in other ways, and we do not need to assume the completeness condition the Eq. (5).
  2. I have rewritten some sentences in the Sec.7 to be more prudent about how our numerical results are supportive of a MIPT. Thanks to the referee for reminding me of this.
  3. In this revision, I have added some numerical results and a discussion about the change of the longtime saturation value of entanglement entropy as the total system size increases using the quantum trajectory method. Also, the details of the quantum trajectory method we have used here and more numerical results are included in section E of the supplementary material.
  4. Thanks to the referee for reminding me of the typos, and I have fixed all the typos about MIPT.

Attachment:

SA_trajectory.pdf

---

## Editorial Decision

resubmitted